# Methodological Aspects of a Comprehensive Analysis of the Fixed Capital of Machine Building Enterprises

**Natalya V. Arsenieva \*, Lyudmila M. Putyatina and Natalia V. Tarasova**

Moscow Aviation Institute, National Research University, Volokolamskoe Highway 4, 125993 Moscow, Russia; Kaf501@mai.ru (L.M.P.); kaf508@mai.ru (N.V.T.)

\*  Correspondence: natars2002@mail.ru

**Abstract:** The article examines in detail the main areas and methodological features of a comprehensive analysis of movement and use of fixed capital of machine building enterprises. The fixed capital of enterprises is particularly important in the process of economic activity and it requires considerable investment for development and improvement. The purpose of this study is to develop a comprehensive approach to the quantitative assessment of change in the fixed assets of enterprises, their structure, and efficiency of use.

**Keywords:** fixed capital; fixed production capital; production capacity; efficiency of the fixed capital use; active and passive part of capital; capital productivity; L23; E22

## 1. Introduction

Machine building enterprises today are the basis for the development of the entire industry of a country. They have a significant physical infrastructure of production, a great variety of machines, equipment, and special tools.

The fixed capital of enterprises is the basis of the material and technical base of production and largely determines its capabilities for innovative renewal of all production and economic activities, diversification of production, and ensuring economic growth.

In the conditions of the current crisis, enterprises are forced to: correct previously developed plans for the production and sale of products, focusing on changes in the market situation; reduce the cost of purchasing new equipment; use limited investments in production development, etc.

This raises an acute economic problem involving the most efficient use of existing fixed assets, their renewal, and their increase, mainly in the most promising and progressive types of production.

In this regard, a comprehensive analysis allows for considering in detail the bottlenecks in the management of fixed assets and justifying an increase in the efficiency of its use in the future in order to increase the quantity and quality of products.

The most rational and well-grounded decisions in the field of fixed capital management of an enterprise allow it to have an undeniable advantage in the investment funds market.

The state and development of fixed capital of enterprises determines the ability to pursue a prospective innovative policy on a wide range of innovations, to develop new products and technologies, to diversify production to ensure market stability, and to develop the full use of production capacities (Arseneva et al. 2015).

The main features of the fixed production assets of the enterprise are as follows: participation in the production process for a long period of time; gradual transfer of its value to the prime cost of products; gradual restoration of its value as it depreciates (Burkaltseva et al. 2019).

The fixed capital of large industrial enterprises is divided into fixed production capital (FPC) and fixed non-production capital (FNC). FPC determines the physical infrastructure of production,

while FNC is the additional property owned by the enterprise, but not participating in the production processes (Golov et al. 2019).

FPC is understood as the means of labor of an enterprise, which is expressed in value terms and divided into the following parts:

- An active part, including the means of labor directly involved in the production process (machines, equipment, monitoring equipment, etc.);
- A passive part, providing production conditions and fulfilling auxiliary functions (buildings, facilities, communications, etc.).

Since the main purpose of the study is the production sphere of enterprises, FPC is the one that is considered substantively.

## 2. Methodology

In modern conditions, a comprehensive analysis of the FPC of an enterprise should be conducted in the following areas (Figure 1):

1. Analysis of the volume, dynamics, and structure of the fixed assets, which gives an idea of the size of the enterprise, the level of development of its technological base, and the rationality of the distribution of funds between the individual components of the FPC (Zheltenkov et al. 2017);
2. Analysis of the efficiency of the FPC use, which shows the impact from the use of the fixed assets of the enterprise, the level of reserves formed for objective and subjective reasons, and the possibility of increasing the rationality of its use (Poelueva and Soldatkina 2019);
3. Analysis of production capacity and the level of its use, which gives an idea of the performance of the enterprise as a whole, the possibilities of quantitative growth for certain types of products, as well as participation in the competition by increasing the market share (Potolova 2019).

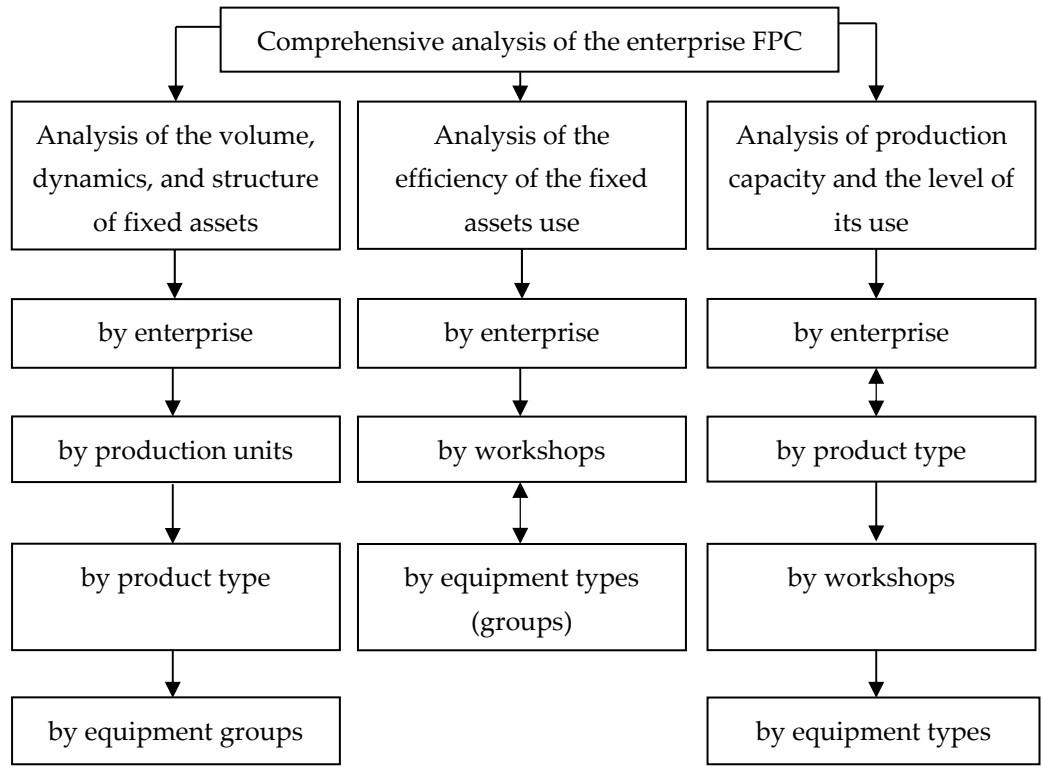

**Figure 1.** Fixed production capital (FPC) analysis for a comprehensive assessment by areas.

The value of the proposed methodological proposals is in the possibility of unlimited increase or reduction of the considered areas of analysis and indicators, depending on the goals and objectives of the study.

Detailed analytical work with this information allows expanding the range of its use for solving various economic problems:

- ranking the renovation of fixed assets of the enterprise;
- development of measures to improve the structure of fixed assets;
- search for reserves to increase the efficiency of using fixed assets by type of product and areas of activity;
- forecasting the growth of manufactured products due to an increase in the efficiency of their use and (or) changes in the structure of fixed assets;
- increasing the profit and profitability of the enterprise due to the improvement of the equipment park and its renewal, etc.

Such analysis may be conducted depending on the objectives: by enterprise, by production units, by types of homogeneous products, and by equipment groups.

The following indicators are used for this analysis (Zheltenkov and Suzeva 2016):

Average annual value of the FPC ($C_{\text{fix}}^{\text{av}}$).

The constant movement of the production capital during the year through purchasing new means of labor, as well as the disposal of obsolete and unused ones, leads to the necessity of calculating the average annual value of the FPC. This indicator can be calculated as follows:

$$C_{\text{fix}}^{\text{av}} = C_{\text{fix}}^{\text{BOY}} + C_{\text{fix}}^{\text{int}} \times \frac{t_1}{12} - C_{\text{fix}}^{\text{ret}} \times \frac{t_2}{12}, \tag{1}$$

where $C_{\text{fix}}^{\text{int}}$ $C_{\text{fix}}^{\text{ret}}$ are the value of the introduced and retired FPC, respectively (RUB); $t_1$ is the number of months during which the introduced FPC worked throughout the year (months); $t_2$ is the number of months during which the retired FPC did not work throughout the year (months).

In the case of conversion of the FPC value into the replacement value of the current period at the beginning of the year (taking into account inflation and other factors), the average annual value is determined as follows:

$$C_{\text{fix}}^{\text{av}} = \sum_{i=1}^{m} C_{\text{fix}_i}^{\text{BOY}} \times I_i + C_{\text{fix}}^{\text{int}} \times \frac{t_1}{12} - C_{\text{fix}}^{\text{ret}} \times \frac{t_2}{12}, \tag{2}$$

where $C_{\text{fix}_i}^{\text{BOY}}$ is the book value of the FPC of the *i*-type at the beginning of the year (RUB); $I_i$ is the price index (conversion rate) of the FPC of the *i*-type.

The increase in the average annual value of the FPC over time characterizes the increase in the production capacity of the enterprise as a whole and its technical equipment, except for in situations when the increase in value occurred only through inflationary recalculation.

The growth rate of the FPC ($R_{C_{\text{fix}}}$) characterizes the level of its change over time. This indicator is calculated as follows:

$$R_{C_{\text{fix}}} = \frac{C_{\text{fix}}^{\text{pl}}}{C_{\text{fix}}^{\text{b}}} \times 100\%, \tag{3}$$

where $C_{\text{fix}}^{\text{pl}}$, $C_{\text{fix}}^{\text{b}}$ are the value of the FPC, respectively, in the planning and base periods (RUB).

The higher the FPC growth rate over time, the more actively the enterprise develops in terms of the development through labor.

The rate of the FPC renewal ($R_{\text{ren}}$) is defined as follows:

$$R_{\text{ren}} = \frac{C_{\text{fix}}^{\text{int}}}{C_{\text{fix}}^{\text{BOY}}}. \tag{4}$$

An increase in this indicator over time means an increase in the investment activity of the enterprise and its technical re-equipment. The value of the introduced capital is taken as a cumulative result at the end of year.

The rate of the FPC retirement ($R_{\text{ret}}$) is defined as follows:

$$R_{\text{ret}} = \frac{C_{\text{fix}}^{\text{ret}}}{C_{\text{fix}}^{\text{BOY}}}. \tag{5}$$

This indicator should be relatively constant and small over time and should characterize a gradual and systematic retirement of the fixed assets (that are outdated or unused due to a change in technological processes) (Allen 2009). The value of the output capital is taken as cumulative result at the end of the year. However, an increase in this indicator over time through active technical re-equipment is possible, which is also regarded as a positive factor from the point of view of the long-term perspective of the enterprise development (Novikov 2018).

In any case, $F_{\text{ret}}$ should not exceed $F_{\text{ren}}$. There is a difference between their values, since the capital is introduced at initial value and retired at residual value. In the context of developing production, the introduction of new means of labor should occur much more intensively than their retirement.

The rate of the FPC increase ($R_{\Delta\text{fix}}$) is calculated as follows:

$$R_{\Delta\text{fix}} = \frac{C_{\text{fix}}^{\text{int}} - C_{\text{fix}}^{\text{ret}}}{C_{\text{fix}}^{\text{BOY}}}. \tag{6}$$

The increase in this indicator over time characterizes the increase in the enterprise property and active renewal of assets.

The proportion of the active part of the FPC ($P_{\text{fix}}^{\text{A}}$) is defined as follows:

$$P_{\text{fix}}^{\text{A}} = \frac{C_{\text{fix}}^{\text{A}}}{C_{\text{fix}}^{\text{EOY}}}, \tag{7}$$

where $C_{\text{fix}}^{\text{A}}$ is the total value of the active part of the FPC at the end of the year (RUB).

The increase in the indicator $P_{\text{fix}}^{\text{A}}$ over time determines the overall improvement in the structure of the FPC.

The depreciation of the fixed assets ($D_{\text{fix}}$) determines the level of novelty of the enterprise FPC. It is calculated as follows:

$$D_{\text{fix}} = \frac{S_{\Sigma}}{C_{\text{fix}}} \times 100\%, \tag{8}$$

where $S_{\Sigma}$ is the total sinking fund of the enterprise at the end of the year (RUB).

The lower the level of wear of the fixed assets, the greater the production capabilities of the enterprise (Galkina 2019).

FPC has the main groups shown in Table 1.

The structure of the FPC of the enterprise is understood as the percentage ratio between the considered groups of fixed production assets. The total amount of the FPC is taken as 100%.

The dynamics of the FPC structure on the example of a machine building enterprise are given in Table 1.

Such types of the fixed assets as buildings, facilities, and transfer devices refer to the passive part. However, the division is largely relative, as they create the necessary conditions for all production processes. Reducing the share of the passive part of the fixed assets is generally achieved by the most rational and efficient use of production space of the enterprise, standard projects for the creation of new production capacities, and the cheapest materials used in the construction and repair of buildings, facilities, etc. (Putyatina and Rodionov 2006).

**Table 1.** Dynamics of the FPC structure of the enterprise.

| FPC Groups | Percentage Ratio by Year | | | Active/Passive | General Trend |
|---|---|---|---|---|---|
| | Base | 1st | 2nd | | |
| 1. Buildings | 10 | 8 | 6 | Passive | relatively constant (+) * |
| 2. Facilities | 5 | 4 | 3 | Passive | relatively constant (+) ** |
| 3. Transfer devices | 5 | 4 | 4 | Passive | relatively constant (+) ** |
| 4. Power equipment | 5 | 3 | 2 | Passive | relatively constant (+) ** |
| 5. Working machinery and equipment | 40 | 46 | 48 | Active | ↑(+) |
| 6. Measuring and regulating equipment | 10 | 6 | 2 | Active | ↑(+) |
| 7. Tools | 4 | 6 | 8 | Active | ↑(+) |
| 8. Means of transport | 7 | 8 | 12 | Passive | depends on the needs |
| 9. Computing machinery | 8 | 10 | 13 | Active/Passive | ↑A (+) |
| 10. Production implements | 6 | 5 | 2 | Active/Passive | ↑A (+) |
| Total FPC | 100 | 100 | 100 | - | - |
| FPC increase | 100 | 110 | 117 | - | ↑(+) |
| Proportion of the active part of the FPC | 60 | 63 | 60 | - | ↑(+) |

\* Except for the prospective expansion of production capacity; ** Except for technical re-equipment of production.

An important point in the analysis of the fixed assets is the fact that the structure itself does not directly reflect the increase in the value of the FPC over the considered period of time.

As a rule, all changes in the structure of the FPC are not incidental. They are always associated with certain production circumstances, such as planned or forced changes in the specialization of the enterprise, fast development of new competitive products, failures of existing equipment, etc. (Ivaniuk and Aryshev 2016).

The analysis of structural shifts requires their quantifying, identifying a set of reasons for positive and negative trends, as well as determining the prospects of the changes in terms of enterprise development.

The main factors affecting the structure of the FPC are as follows:

1. Specialization of the enterprise, i.e., the nomenclature of products and work performed;
2. Production volumes (size of the enterprise);
3. Level of production cooperation;
4. Level of mechanization and automation;
5. Location of the enterprise (in regard to sources of raw materials, fuel, heat and energy supply, etc.);
6. Climatic conditions, etc.

Improving the structure of the fixed assets is possible through the following:

- timely (planned) renewal and modernization of existing equipment;
- efficient use of production space through the rational arranging of the equipment;
- improving the structure of the equipment fleet with the help of new high-performance and accurate equipment and the means of mechanization and automation of the main and auxiliary production processes;
- use of standard projects in expanding production capacities;
- timely liquidation of unused equipment (sale or write-off on the expiry of the service life), etc.

The results of the analysis of the FPC volume, dynamics, and structure are as follows:

1. Determination of the general direction of change in the FPC value of the enterprise (increase, decrease, and relative constancy);
2. Quantitative assessment of changes in the FPC (by the growth rate);
3. Determination of the level of the FPC renewal (through the introduction and retirement of the fixed assets);



4.  Assessment of the level of the FPC depreciation and its regularity over time;
5.  General assessment of the FPC structure (improvement, deterioration, and relative constancy), as well as the change in the share of the most important components of its active part.

If the FPC analysis is conducted on production units, then indicators, calculation formulas, and conclusions are made similarly.

All indicators of the efficiency of the FPC use can be divided into two groups:

1.  Indicators of the intensive and extensive use of the FPC (in particular, equipment), which differ from each other in that the former assess their use on the basis of productivity, and the latter assess it on the basis of the time used;
2.  Integral indicators of the efficiency of the use of the FPC, defining the combination of all factors affecting the efficiency of its use.

The indicators of the 1st group include:

The rate of the intensive equipment use ($R_{int}$), which is calculated by the following formula:

$$R_{int} = \frac{O_a}{O_n},\qquad(9)$$

where $O_a$ is the actual products output per unit of time (a day, month, quarter, or year); $O_n$ is technically justified products output per unit of time (determined on the basis of equipment passport data).

This indicator can be calculated both for specific types of equipment and for groups, sections, workshops, etc. It characterizes the efficiency of equipment use in accordance with its technical capabilities.

The following indicators are used as indicators characterizing the level of use of the FPC (mainly the active part) over time:

The rate of the extensive equipment use ($R_{ext}$), which is calculated as follows:

$$R_{ext} = \frac{t_a}{t_{pl}},\qquad(10)$$

where $t_a$ is the actual equipment work time (hours); $t_{pl}$ is the planned fund of equipment work time (hours).

The value of this rate can be calculated both for specific types of equipment and for equipment groups, workshops, sections, etc.

The equipment shift rate ($R_{sh}$) shows the average number of shifts that each unit of equipment of the enterprise works annually. Generally, the shift rate is calculated as follows:

$$R_{sh} = \frac{n_{ms}}{N_{inst} \times D_w},\qquad(11)$$

where $n_{ms}$ is the total number of machine shifts worked at the enterprise (in the workshop) per month (year); $N_{inst}$ is the number of the equipment installed at the enterprise (in the workshop); $D_w$ is the number of working days in a month (year). This is an important indicator of increasing production efficiency, which does not require additional investment costs to increase the fixed assets.

An increase in this indicator over time means a prospective growth in output volumes. The equipment load rate ($R_{load}$), as well as the previous rates, characterizes the level of equipment use over time. It is determined within the workshop or section by the following formula:

$$R_{load} = \frac{\sum_{i=1}^{k}(T_i \times N_i)}{\sum_{j=1}^{m} T_{pl_j}},\qquad(12)$$

where $T_i$ is the labor intensity of the work (product) of the *i*-type performed in a given workshop (section) in a month (year) (hours or standard hours); $k$ is the number of names of work types performed

in the workshop (section) in the considered period of time (month, year); $N_i$ is the volume of products (work) of the *i*-type made in the workshop in a month (year) (pcs.); $T_{pl_j}$ is the planned fund of work time of the equipment unit of the *j*-type in a month (year) (hours); *m* is the number of equipment units installed in the workshop (section) during the month (year).

The rate of use of the shift mode of the equipment work ($R_{sh.w}$) is calculated as follows:

$$R_{sh.w} = \frac{R_{sh}}{t_{sh}}, \tag{13}$$

where $R_{sh}$ is the shift rate of the equipment at the enterprise (in the workshop) reached in a given period; $t_{sh}$ is the shift duration at the enterprise (in the workshop) (hours).

The rate of use of the enterprise production capacity (by labor intensity) ($R_{use}$) is determined similarly with the equipment load rate, but it takes into account the entire nomenclature of products of the enterprise, which goes through several workshops of the enterprise during production

$$R_{use} = \frac{\sum_{i=1}^{l}(T_i \times N_i)}{\sum_{j=1}^{g} T_{jpl}}, \tag{14}$$

where $T_i$ is the labor intensity of the *i*-type product in the accounting year (hours or standard hours); $N_i$ is the volume of products of the *i*-type sold (produced) by the enterprise in a year (pcs.); *l* is the nomenclature of products in a year; $T_{jpl}$ is the planned fund of the *j*-type equipment work time in a year (hours); *g* is the number of the equipment units installed and working at the enterprise during the year.

The dynamics of increase in the rate of use of the enterprise production capacity characterizes its production activity, increase in the efficiency of the fixed assets use, and in the level of organization of production.

In the analysis of the equipment use, it is important to compare its extensive load with the intensive one, since the equipment may actually idle or produce low-quality products. With the efficient use of equipment, an increase in the indicators of the extensive equipment use should be accompanied by increased indicators of its intensive use.

For a generalized assessment, indicators determining the integral assessment of the efficiency of the FPC use for the enterprise as a whole are of the greatest importance. Among these indicators, the most commonly used ones are the following:

1.　rate of integral use of the equipment of the enterprise ($R_{int.eqip}$), which reflects its operation in time and productivity (capacity). It is calculated as follows:

$$R_{int.eqip} = R_{ext} \times R_{int}, \tag{15}$$

where $R_{ext}$, $R_{int}$ are the rates of the extensive and intensive equipment use, respectively.

The dynamics of the increase in this indicator determine the general process of improving the organization of production at the enterprise, even without increasing its production capacity.

2.　capital productivity ($C_{prod}$), the economic meaning of which is determined by the amount of roubles of the sold products of the enterprise per 1 rouble of the average annual value of the fixed assets, i.e.,

$$C_{prod} = \frac{Q_s}{C_{fix}^{av}} \text{ (RUB/RUB)}, \tag{16}$$

where $Q_s$ is the annual volume of sold products of the enterprise at current market prices (RUB) and $C_{fix}^{av}$ is the average annual value of the fixed assets of the enterprise (RUB).

The dynamics of an increase in capital productivity indicate general improvement of the efficiency of the FPC use. In the context of multinomenclature production, the annual volume of sold products is determined as follows:

$$Q_s = \sum_{i=1}^{k} (P_i \times N_i),\tag{17}$$

where $P_i$ is the market price (selling price without value added tax) of the $i$-type product of the enterprise (RUB/pieces) in the current year; $N_i$ is the volume of sales of the $i$-type product in the current year (pcs.); $k$ is the nomenclature of the products produced (sold) by the enterprise in the current year.

Growth factors of sold products and, as a result, of capital productivity can be purely price (increase in prices in specific conditions), volumetric (increase in actual production volumes in physical terms), and mixed ones (volumetric, price, and structural by nomenclature). The increase in capital productivity exclusively by price factors and the retirement of the fixed assets without their replenishment for any reason cannot be considered a positive trend in increasing the efficiency of the FPC use.

Therefore, the dynamics of changes in capital productivity, determined by calculation, requires additional factor analysis.

Capital intensity ($C_{int}$) is the reciprocal of the capital productivity, which characterizes the amount of roubles of the fixed assets per 1 rouble of the sold product, i.e.,

$$C_{int} = \frac{C_{fix}^{av}}{Q_s} \text{ (RUB/RUB)}.\tag{18}$$

The trend of decrease in the indicator over time determines an increase in the efficiency of the FPC use.

The capital-labor ratio ($C_{rat}$) characterizes the level of provision of the workers with necessary means of labor. It is determined both on the basis of the total number of employees (all industrial production personnel) ($C_{1IPP}$) at the enterprise and the direct workers ($C_{1wkr}$). These indicators, respectively, are calculated as follows:

$$C_{1IPP} = \frac{C_{fix}^{av}}{N_{IPP}^{av}} \text{ (RUB/pers.)};\tag{19}$$

$$C_{1wkr} = \frac{C_{fix}^{av}}{N_{wkr}^{av}} \text{ (RUB/pers.)},\tag{20}$$

where $N_{IPP}^{av}$ and $N_{wkr}^{av}$ are the average number of the industrial production personnel and workers, respectively (pers.).

The dynamics of the increase in the indicator determines the increase in the level of provision with labor resources. However, the growth of the capital-labor ratio should be adequate to the growth of capital productivity at the enterprise. Otherwise, additional equipment for production and personnel will only increase production costs and the prime cost of the products.

Output of the products per 1 m$^2$ of production space ($O_{1m^2}$) determines the efficiency of the use of industrial buildings and facilities. The economic meaning of the indicator is the amount of roubles of the products sold in a year per 1 m$^2$ of the production space. It is calculated as follows:

$$O_{1m^2} = \frac{Q_s}{S_{pr}} \text{ (RUB/m}^2\text{)},\tag{21}$$

where $S_{pr}$ is the production space of the enterprise (m$^2$).

The dynamics of the increase in the indicator reflects the most efficient use of the passive part of fixed assets by the enterprise.

The profitability of the FPC ($P_{\text{fix}}$) characterizes the overall efficiency of the economic activity of the enterprise, in which the fixed assets are used. It is calculated as follows:

$$P_{\text{fix}} = \frac{PR_{\text{BS}}}{C_{\text{fix}}^{\text{av}}} \times 100\%, \tag{22}$$

where $PR_{\text{BS}}$ is the balance sheet profit made by the enterprise as a result of financial and economic activities in a year (RUB).

Since the balance sheet profit may contain purely financial income (non-operating operations), upon which the FPC has a very indirect bearing, it is more appropriate to determine the profitability indicator as follows:

$$P_{\text{fix}}^* = \frac{PR_{\text{P}}}{C_{\text{fix}}^{\text{av}}} \times 100\%, \tag{23}$$

where $PR_{\text{P}}$ is the profit from commodity production of the enterprise in a year (RUB).

The increase in the indicator over time shows an increase in the efficiency of use of both active and passive part of the fixed assets.

Return on production assets ($R_a$) determines the level of increase in the balance sheet profit ($\Delta Pr_{bs}$) in relation to the increase in the FPC ($\Delta C_{\text{fix}}^{\text{av}}$) of the enterprise, i.e.,

$$R_a = \frac{\Delta Pr_{BS}}{\Delta C_{\text{fix}}^{\text{av}}} \times 100\% = \frac{Pr_{\text{BS}}^{\text{pl}} - Pr_{\text{BS}}^{\text{base}}}{\left( C_{\text{fix}}^{\text{av(pl)}} - C_{\text{fix}}^{\text{av(base)}} \right)} \times 100\%, \tag{24}$$

where $Pr_{\text{BS}}^{\text{pl}}$ and $Pr_{\text{BS}}^{\text{base}}$ are the balance sheet profit of the enterprise in the planning and base periods, respectively (RUB); $C_{\text{fix}}^{\text{av(pl)}}$ and $C_{\text{fix}}^{\text{av(base)}}$ are the average annual value of the FPC of the enterprise in the planning and base periods, respectively (RUB).

The increase in the indicator over time confirms the efficiency of the increase in the fixed assets of the enterprise.

## 3. Results and Discussion

To illustrate the considered methodological provisions, we use the indicators of two enterprises ($E_1$ and $E_2$) for the main areas and a limited list of indicators for comparison.

Table 2 shows initial and calculation data for two compared competing enterprises in the industry $E_1$ and $E_2$. The development and use of the fixed production assets of the enterprises was assessed by using three years: the base, first, and second one.

The enterprises were compared by using two methods: at the qualitative level, when the advantage of one enterprise over another for each indicator is assessed (+); at the qualitative level, using the method of expert ratings in points.

**Table 2.** Initial and calculation data for the analysis of the FPC of the studied enterprises.

| Enterprises | $E_1$ | | | | $E_2$ | | | |
|---|---|---|---|---|---|---|---|---|
| Years Indicators | Base | 1st | 2nd | Assessment | Base | 1st | 2nd | Assessment |
| **Initial Indicators** | - | - | - | - | - | - | - | - |
| 1. $Q_s$ (millions of RUB) | 2880 | 3500 | 4200 | (+) | 3600 | 3900 | 4200 | (+) |
| 2. $C_{\text{fix}}$ (millions of RUB) | 424 | 427 | 462 | (−) | 379 | 398 | 350 | (+) |
| 3. $N_{\text{IPP}}$ (pers.) | 1200 | 1150 | 1020 | (+) | 1300 | 1400 | 1500 | (−) |
| 4. $Pr_{\text{BS}}$ (millions of RUB) | 288 | 360 | 540 | (+) | 250 | 320 | 350 | (−) |
| **Calculation Data** | - | - | - | - | - | - | - | - |
| 5. $R_{Q_s}$ (%) | - | 121.5 | 120 | (+) | - | 108.3 | 107.7 | (−) |
| 6. $R_{\text{Cfix}}$ (%) | $R_{C_{\text{fix}}}$ | 100.7 | 108.2 | (+) | - | 105.0 | 88.0 | (−) |
| 7. $R_{\text{PRbs}}$ (%) | - | 127.1 | 150 | (+) | - | 128 | 109.4 | (−) |

**Table 2.** *Cont.*

| Enterprises | $E_1$ | | | | $E_2$ | | | |
|---|---|---|---|---|---|---|---|---|
| Years Indicators | Base | 1st | 2nd | Assessment | Base | 1st | 2nd | Assessment |
| **Calculation Indicators of the Efficiency of the FPC Use** | - | - | - | - | - | - | - | - |
| 8. $C_{prod}$ (RUB/RUB) | 6.8 | 8.2 | 9.1 | (−) | 9.5 | 9.8 | 12.0 | (+) |
| 9. $C_{rat}$ (thousands of RUB/pers.) | 2400 | 3043 | 4118 | (+) | 2769 | 2786 | 2666 | (−) |
| 10. $P_{fix}$ (%) | 67.9 | 83.4 | 116.9 | (+) | 66.0 | 80.4 | 105.1 | (−) |
| 11. $I$ (%) | 68 | 72 | 85 | (−) | 50 | 58 | 64 | (+) |
| 12. $S_a$ (%) | 42 | 48 | 50 | (−) | 55 | 62 | 68 | (+) |
| 13. $R_{use}$ (dimensionless quantity) | 0.6 | 0.7 | 0.75 | (+) | 0.55 | 0.58 | 0.6 | (−) |

The analysis of the initial data of the compared enterprises (Table 2) showed the following:

1.  Both enterprises were actively increasing the volume of sales ($Q_s$) in relation to the base year in the considered period of time and made identical revenue of 4200 million of RUB at the end of the period. This means that the activity of both enterprises can be assessed positively (+);

2.  However, the enterprise $E_1$ had a higher growth rate of sales: 121.5% of sales occurred in the 1st year in relation to the base one and 120% in the 2nd year in relation to the 1st one. Therefore, according to this indicator, priority (+) can be given to the enterprise $E_1$;

3.  The enterprise $E_2$ has a lower average annual value of the fixed assets ($C_{fix}$) at the end of the period. However, taking into account the identical revenue, the positive trend of the change in $C_{fix}$ (+) should be recognized for the enterprise $E_2$;

4.  The enterprise $E_1$ consistently increases $C_{fix}$ (by 9% in general), and the enterprise $E_2$ reduces it by 12%. This situation is rather difficult, since the increase in the fixed assets does not mean anything in terms of the efficiency of their use. However, (+) should be formally given to the enterprise $E_1$;

5.  The number of personnel ($N_{IPP}$) at the compared enterprises changes in different ways. At the enterprise $E_1$, it decreases in general by 15% in relation to the base period, and at the enterprise $E_2$, it increases by 15%. If the enterprise increases the production volumes while reducing the number of personnel, it indicates that it seeks to improve the efficiency of the use of labor resources, and in this case the positive trend (+) should be recognized for the enterprise $E_1$;

6.  The enterprise $E_1$ reached a higher value of the balance sheet profit ($PR_{BS}$) at the end of the period (540 million of RUB compared to 350 million of RUB), which provides it with another advantage (+);

7.  The growth rate of the balance sheet profit of the enterprise $E_2$ is also higher (127.1% and 150%, respectively, in the periods), which characterizes higher efficiency of its commercial activities (+).

In general, the development of the enterprise $E_1$ should be recognized as more dynamic and proportional (according to the considered indicators, it received 6 positive ratings (+) of 7 indicators, while the enterprise $E_2$ has only 2 positive assessments (+) of 7 indicators.

The analysis of the dynamics, structure, and efficiency of the FPC use, taking into account the use of the production capacity of the compared enterprises, showed the following:

1.  The value of the capital productivity ($C_{prod}$) of the enterprise $E_2$ is higher both at the beginning and at the end of the period; therefore, its efficiency of the fixed assets use is also higher (+);

2.  The capital-labor ratio ($C_{rat}$) is higher at the enterprise $E_1$, which means the higher level of provision of the personnel with means of labor is a positive factor (+);

3.  The profitability of the FPC ($P_{fix}$) for the entire period is also higher at the enterprise $E_1$;

4.  The depreciation of the fixed assets is lower at the enterprise $E_2$, which gives it a certain advantage (+), since it means that its fixed assets are more operational;

5.  The share of the active part $S_a$ of the fixed assets is higher for the enterprise $E_2$, which characterizes a more rational structure of the fixed assets (+);

6.　The rate of use of the enterprise production capacity at the enterprise $E_1$ is higher during the entire considered period of time, which indicates the most efficient work of the enterprise and higher level of organization of production (+).

In general, the condition, structure, and efficiency of use of the fixed assets at the enterprise $E_1$ is higher in comparison with the enterprise $E_2$ (at the enterprise $E_1$ 9 indicators of 13 have higher positive results (+) in Table 2, while at the enterprise $E_2$ only 5 indicators of 13 have higher positive results (+).

When carrying out such analytical calculations, it is possible to increase the considered indicators, for example, such as:

- level of complexity of the products;
- rate of growth of fixed assets;
- growth rates of the most promising equipment;
- rate of growth of workers;
- level of automation and mechanization of production;
- level of use of digital economy methods in the production process and many others.

Practice shows that an excessive amount of analytical data very often complicates the objectivity of the results and conclusions. In this regard, the assigned analytical tasks should have the necessary and sufficient number of indicators.

Now we are going to provide a rating assessment of the main indicators of the condition, structure, and efficiency of the fixed assets use of the analyzed enterprises. Indicators for comparing the enterprises over time are given in Table 3.

**Table 3.** Rating assessment of the efficiency indicators of the fixed capital use of the studied enterprises ($E_1$ and $E_2$) (in points).

| Enterprises Years Indicators | Maximum Points | $E_1$ | | $E_2$ | |
| --- | --- | --- | --- | --- | --- |
| | | 1st Year | 2nd Year | 1st Year | 2nd Year |
| 1. $C_{prod}$ (points) | 10.00 | 8.40 | 7.60 | 10.00 | 10.00 |
| 2. $P_{fix}$ (points) | 8.00 | 8.00 | 8.00 | 7.60 | 7.20 |
| 3. $S_a$ (points) | 7.00 | 5.40 | 5.10 | 7.00 | 7.00 |
| 4. $I$ (points) | 6.00 | 4.80 | 4.50 | 6.00 | 6.00 |
| 5. $C_{rat}$ (points) | 5.00 | 5.00 | 5.00 | 3.20 | 3.20 |
| 6. $R_{use}$ (points) | 4.00 | 3.70 | 4.00 | 3.14 | 3.20 |
| Total | - | 35.30 | 34.20 | 36.94 | 36.60 |

We conducted a rating comparison of the enterprises on the basis of the efficiency of the FPC use according to the following methodology:

1.　We selected indicators for comparison and determined their priority for a comprehensive assessment on the basis of the maximum number of points (10);
2.　The maximum value of the indicator for an enterprise was equated to the maximum number of points, if a higher value of the indicator corresponded to higher efficiency. If the minimum value of the indicator corresponded to higher efficiency, it was given the maximum points, for instance, in the context of depreciation of the fixed assets;
3.　The actual value of points, which corresponds to other values of the indicators, was determined by the direct or inverse proportion;
4.　All points of the enterprises for the corresponding periods of time were summarized. The maximum amount of points corresponded to a higher level of efficiency of use of the fixed assets in a comprehensive assessment.

A rating comparison showed that in both in the 1st and the 2nd year, the enterprise E2 in a comprehensive assessment had a slight advantage in the structure and the efficiency of use of the FPC, which is about 4.6% (36.94/35.3 × 100%) in the 1st year and about 7% (36.6/34.2 × 100%) in the 2nd year.

However, it should be noted that the enterprise $E_2$ had significant reserves and opportunities for increasing the efficiency of use of the production capacity with a corresponding increase in the FPC of the enterprise.

In spite of the slight advantage of the enterprise $E_2$ in terms of the efficiency of the FPC use in a comprehensive assessment, it is necessary to recognize the development of the enterprise $E_1$ as the most balanced in the whole range of considered indicators.

## 4. Conclusions

Today, eliminating the sad consequences of the pandemic and the forced downtime of production, enterprises should pay special attention to the most efficient use of fixed capital as the basis for their further innovative development. As was mentioned earlier, fixed capital of enterprises determines their capabilities for innovative renewal of all production and economic activities, diversification of production, and ensuring economic growth. Changes in the external environment led to the fact that enterprises were forced to constantly reduce their costs, including by refusing to upgrade production capacities, adjust the volumes, terms, and conditions of production, and seek reserves in internal production activities.

In this regard, comprehensive analysis allows for considering the bottlenecks in the management of fixed assets in detail and justifying an increase in the efficiency of its use in the future in order to increase the quantity and quality of products.

The most rational and well-grounded decisions in the field of fixed capital management of an enterprise allow it to have an undeniable advantage in the investment funds market.

The value of the proposed methodological and methodological proposals lies in the possibility of unlimited increase or reduction of the considered areas of analysis and indicators, depending on the goals and objectives of the study.

Detailed analytical work with this information allows expanding the range of its use for solving various economic problems.

A comprehensive analysis of the FPC of the enterprise allows the following:

1. Considering its change over time to determine positive and negative development trends.
2. Analyzing the structure of the capital in order to determine its rationality.
3. Calculating the efficiency of the capital use for the development of measures to increase it in the future.
4. Identifying reserves for increasing production capacity for production output and other aspects.

The study is distinguished by novelty in terms of the volumetric approach to the FPC, which is considered in the dynamics of its development and allows finding certain disparities and miscalculations to correct them in the future.

Efficient FPC management is an important component in the development of the physical infrastructure of the production, which, in turn, provides the enterprise with the opportunity to increase not only the efficiency of use of the particular type of resources, but also the overall efficiency of work.

An increase in production capacity of the enterprise and the level of its use means an increase in output, revenue, and profit, which is an important source of self-financing for the development and improvement of personnel welfare.

The balanced development of the FPC with limited resources means the excess of the growth rate of their efficiency over the growth rate of the capital itself. The optimal and balanced growth of the FPC determines the economic growth of the economic entities in general and the industrial economy in particular.

Practical calculations showed the relative simplicity of the studies and the efficiency of the obtained results and inferences.

**Author Contributions:** Conceptualization, N.V.A. and L.M.P.; methodology, N.V.T.; software, N.V.T.; validation, N.V.A., L.M.P. and N.V.T.; formal analysis, N.V.T.; investigation, N.V.A.; resources, N.V.A.; data curation, N.V.A., L.M.P. and N.V.T.; writing—original draft preparation, L.M.P.; writing—review and editing, N.V.A., L.M.P. and N.V.T.; visualization, N.V.A.; supervision, L.M.P.; project administration, N.V.A.; funding acquisition, N.V.T. All authors have read and agree to the published version of the manuscript.

**Funding:** This research received no external funding.

**Conflicts of Interest:** The authors declare no conflict of interest.

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
