# Peer review of "Methodological Aspects of a Comprehensive Analysis of the Fixed Capital of Machine Building Enterprises"

_economies, doi:10.3390/economies8030073_

Round 1
Reviewer 1 Report
This article is interesting but I suggest to stress more the economic relevance of this analysis. We are in a period of intense technology change. Authors must explain at which extent this methodology can help companies to decide for their organization of production with respect to the market strategy they decide to follow. Authors must expand introduction and conclusion to develop the economic implications of a such complex methodological effort, to improve the originality and overall merit of this paper.
Author Response
Dear sirs,
Thank you for the review. Please find attached the comments.
Best regards,
Authors

Reviewer 2 Report
The theme is interesting. Nevertheless, I think it could be improved:
- Introduction - explain the interest of the methodology and the added value of this paper;
- Methodology - should improve the reasoning and presentation of the ratios;
- Results and discussion - enrich the example with the calculation of eventually more indicators and improve the calculation presentation (Table 2);
- Conclusions - better explain the added value of this paper, given the current stage of the investigation.
Author Response

(The authors gave the same response as above.)

Round 2
Reviewer 1 Report
the paper can be accpted. the introductory and final notes clarify the economic content of this complex methodological exercise, pb